# AIME: AI System Optimization via Multiple LLM Evaluators

## Abstract

Text-based AI system optimization typically involves a feedback loop scheme where a *single* LLM generates an evaluation in natural language of the current output to improve the next iteration's output. However, in this work, we empirically demonstrate that for a practical and complex task (code generation) with multiple criteria to evaluate, utilizing only one LLM evaluator tends to let errors in generated code go undetected, thus leading to incorrect evaluations and ultimately suboptimal test case performance. Motivated by this failure case, we assume there exists an optimal evaluation policy that samples an evaluation between response and ground truth. We then theoretically prove that a linear combination of multiple evaluators can approximate this optimal policy. From this insight, we propose **AI system optimization via Multiple LLM Evaluators (AIME)**. AIME is an evaluation protocol that utilizes multiple LLMs that each independently generate an evaluation on separate criteria and then combine them via concatenation. We provide an extensive empirical study showing AIME outperforming baseline methods in code generation tasks, with up to $62\%$ higher error detection rate and up to $16\%$ higher success rate than a single LLM evaluation protocol on LeetCodeHard and HumanEval datasets. We also show that the selection of the number of evaluators and which criteria to utilize is non-trivial as it can impact pact success rate by up to $12\%$.

## 1 Introduction

Pre-trained foundation models, such as Large Language Models (LLMs), have developed rapidly over the recent years (Achiam et al., 2023; Touvron et al., 2023). With these advancements, AI systems have grown in popularity for various tasks such as code generation (Chen et al., 2024; Gulwani, 2010), question-answering (Patel et al., 2024; Wang et al., 2024), mathematical reasoning (Trinh et al., 2024; Song et al., 2024), exploration (Dorbala et al., 2024; 2023; Ren et al., 2024), and information retrieval (Gao et al., 2023) etc. As the application complexity increases, the shift to AI systems containing multiple components such as LLM-based agents and web search (Xiong et al., 2024), will continue (Zaharia et al., 2024; Yuksekgonul et al., 2024). Thus, automatically optimizing these systems, *AI system optimization* (Yuksekgonul et al., 2024), becomes increasingly necessary.

An emerging paradigm is text-based optimization, also known as prompt optimization (Cheng et al., 2023; Wang et al., 2023; Zhou et al., 2022), whereby the natural language input prompt is tuned to generate an optimal output. This method requires no numerical gradient descent updates typical in optimization for machine learning models (Van Der Malsburg, 1986; Hassoun, 1995; Barto, 1992) and is thus appropriate for optimizing AI systems with fixed LLM components. Recently, there has been a growing class of iterative online methods for text-based optimization (Cheng et al., 2024; Yuksekgonul et al., 2024; Shinn et al., 2024), where a *single* LLM generates an evaluation based on the current output to help generate the next iteration's prompt.

While prior art has compared the abilities of a single LLM for evaluations against those of multiple LLMs (Kocmi & Federmann, 2023; Ankner et al., 2024), in AI system optimization literature, there has been a lack of studies questioning the capabilities of using a single LLM evaluator to drive the optimization process. Recently, Yuksekgonul et al. (2024) has viewed the evaluation as a text-based analogy to the objective function for backpropagation (Hinton et al., 2006; Rumelhart et al., 1986)

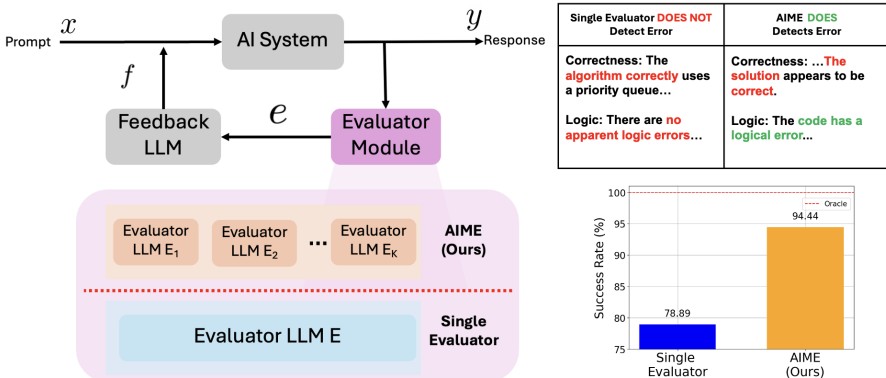

Figure 1: **AI System Optimization Pipeline and Increased Error Detection and Success Rate with AIME-based Evaluation: [LEFT]** Text-based AI system optimization with SoTA framework (Yuksekgonul et al., 2024) using our multiple LLM evaluator approach AIME (orange) and with single-evaluator approach (blue). **[TOP RIGHT]** The single-evaluator approach cannot detect an error in the generated code that fails all test cases. However, one of the evaluators of AIME could because the logical evaluator was independent from the correctness evaluator. **[BOTTOM RIGHT]** AIME-based optimization achieves a $\sim 16\%$ higher success rate than a single-evaluator approach in code generation tasks.

in deep learning optimization. The objective function is a crucial element in optimizing machine learning models (Christiano et al., 2017; Mescheder et al., 2018; Chakraborty et al., 2023; Kingma & Welling, 2014). This importance motivates us to analyze and strengthen the evaluation protocol of state-of-the-art (SoTA) AI system optimization frameworks by addressing a critical research question: *What are the failure cases or tasks of utilizing only one LLM-based evaluator for text-based AI system optimization?*.

For this question, we empirically demonstrate the shortcomings of a single evaluator protocol in judging complex outputs like code based on multiple diverse criteria, such as correctness, readability, and runtime. We emphasize its practical limitations to give optimal evaluation while being instructed to judge based on all criteria simultaneously. Figure 1 illustrates the suboptimality in the practice of an AI system optimization framework with a single-evaluator approach to code generation. Furthermore, by assuming there exists an optimal evaluation policy that in expectation samples the true evaluation between the generated response and ground truth, we also theoretically highlight that the suboptimality gap between a single evaluator and an optimal evaluator is fixed and cannot be reduced given the same output and problem task. With this insight, we then naturally ask the following subsequent query: *Can we develop a principled evaluation method for text-based optimization to handle multiple criteria?* We address this question by assuming there exists an optimal evaluation policy that in expectation samples the true evaluation between the generated response and ground truth. We then theoretically prove that, under a linear additivity assumption, increasing the number of evaluators can reduce the suboptimality gap. We capitalize on this theoretical insight by proposing **AIME: AI** system optimization via **M**ultiple **E**valuators. AIME generates and combines via concatenation independent natural language evaluations from multiple evaluators based on different evaluation instructions. We demonstrate on code generation tasks with LeetCodeHard and HumanEval benchmarks the superior performance of AIME over a single evaluator in code error detection and the success rate of test cases.

Our main contributions are as follows:

- **Novel Evaluation Approach for AI system Optimization:** We propose using multiple LLM-based evaluators and introduce our AIME approach for iterative AI system optimization. We concatenate independent diverse samples from multiple LLM-based evaluation policies to better critique system outputs.

- **Theoretical Motivation for Multiple Evaluators:** We prove that through a linear additivity assumption increasing the number of evaluations can reduce the suboptimatity gap from

an optimal evaluation policy while a single evaluator has a fixed gap. This theoretical result helps justify our formulation for a multiple evaluation-based protocol.

- **Empirical Performance Over Single-Evaluation Approach:** Using popular code generation dataset, LeetCodeHard (Shinn et al., 2024) and HumanEval (Chen et al., 2021), we perform an extensive study showing the superior prowess of AIME with 6 evaluators over single evaluation to detect errors, with AIME achieve up to $62\%$ higher error detection rate than single evaluation. We then show that AIME-based optimization achieves up to a $16\%$ higher success rate on test cases than optimization with only a single evaluator. We also reveal that the choice of the number of evaluators and the combination of criteria to utilize can affect the success rate by up to $12\%$, emphasizing the design of AIME-based optimization is non-trivial. We provide an anonymized code repository. [1]

## 2 TEXT-BASED AI SYSTEM OPTIMIZATION

**Objective Function.** In this section, we now characterize mathematically text-based prompt optimization as a system of LLM-based policies. Let $\pi(\cdot|x)$ be the LLM-based AI system parameterized by fixed LLM-based policy that samples an output response $y \sim \pi(\cdot|x)$ given an input prompt $x \in \mathcal{X}$ from the set of prompts $\mathcal{X}$. We aim to sample a $y \sim \pi(\cdot|x^*)$ by finding an input prompt $x^*$ corresponding to $x$ prompt such that $y$ is closer to the optimal response $y^*$. For code generation, $\pi_\theta$ would be the LLM generator; $x$ would be the input prompt; $y$ is the generated code; and the $y^*$ here would be a code snippet that is a readable, efficient solution to the problem. Mathematically, we can write

$$x^* = \arg\min_{x \in \mathcal{X}} \mathbb{E}_{y \sim \pi_\theta(\cdot|x)}[l(y^*, y)], \tag{1}$$

where $l$ is a loss function to capture the closeness of sampled response $y$ to the ground truth $y^*$.

**Iterative text-based optimization.** Given an initial prompt $x_1$, we perform an iterative text-based optimization method to find $x^*$ as follows. For each iteration $t = 1$ to $T$, we start by (i) sampling $y_t \sim \pi_\theta(\cdot|x_t)$, (ii) evaluate the response $y_t$ to obtain evaluation $e_t = l(y^*, y_t)$, and then finally (iii) generate the next prompt $x_{t+1} \sim \pi(\cdot|y_t, e_t, x_t)$. Recent work by Yuksekgonul et al. (2024) decompose step (iii) into two separate steps and (iii.a) first generate the feedback $f_t \sim \pi(\cdot|y_t, e_t, x_t)$, and then (iii.b) generate the next prompt $x_{t+1} \sim \pi(\cdot|y_t, f_t, x_t)$. For simplicity, we use the same variable $\pi$ for all LLM-based policies because the outputs are dependent on the input variables the policy is conditioned on, so the same LLM model can be utilized. In this paper, we use the same model, GPT-4o, for all steps. However, distinct LLM models can be employed at different steps.

**Challenges.** In an ideal setting, if we had the access to $y^*$ as in supervised learning (Tiwari, 2022), then we can achieve the optimal performance with larger data. However, in practice, they are hard to obtain or simply unknown for many tasks such as code generation (Chen et al., 2024). Therefore, a direct comparison to an optimal output $y^*$ and the resulting calculation of $e$ in step (ii) are both infeasible. Current SoTA work instead *sample* an evaluation $e$ from an evaluation policy conditioned by the response output $y$ and prompt $x$ as $e \sim \pi(\cdot|x, y)$. Let us denote $\pi_e = \pi(\cdot|x, y)$ for notation simplicitiy. Ideally, we would like the evaluation $e$ of $y$ to be $l(y^*, y)$. More specifically, let's assume the existence of an optimal evaluator LLM denoted by $\pi_e^*$, sampling from which will give us samples of the true loss function $l(y^*, y)$.

**Fixed Gap in Evaluation with Single Evaluation Policy from Prior SOTA.** As $\pi_e^*$ is unavailable as discussed before, current SoTA methods sample the evaluation loss from a single evaluator as $e \sim \pi_e$. Now, we know that in the majority of the scenarios $\pi_e$ will not be the true evaluator policy $\pi_e^*$. Thus $e = l(\hat{y}, y)$, where $\hat{y}$ is an implicit approximation of $y^*$ from $\pi_e$. Under this scenario, we define the suboptimality gap in evaluation of prior SOTA as

$$\Delta_{\text{Eva-sub-opt}}^{\pi} = \mathbb{E}_{e^* \sim \pi^*(\cdot|x, y)}[e^*] - \mathbb{E}_{e \sim \pi(\cdot|x, y)}[e] \leq |e|_{\max} d_{\text{TV}}(\pi_e^*(\cdot|x, y), \pi(\cdot|x, y)) \tag{2}$$

where we first expand upon the sub-optimality in evaluation and then upper-bound using the total variation distance (cite ipm). We see that the term $d_{\text{TV}}(\pi_e^*(\cdot|x, y), \pi(\cdot|x, y))$ is fixed and it can't be improved once we have the evaluator $\pi$. This result shows the hardness of a single evaluator

---

[1]Anonymous repository to code: https://anonymous.4open.science/r/aime-4D3E/README.md

reaching $\pi_e^*$ due to this constant gap and it will only reduce if our current LLM evaluator is near-optimal which is not true in majority of the scenarios. Empirically, Figure 1 demonstrates a practical observation where a single evaluator lets code errors go undetected, causing a large suboptimality gap from oracle performance in code generation tasks.

# 3 AIME: AI System Optimization via Multiple LLM Evaluators

Our key idea is to utilize multiple evaluations than single evaluators used in state-of-the-art. The thought that multiple evaluators would work better than one sounds intuitive but a naive introduction of multiple evaluators does not work in practice. We theoretically prove the merit of multiple evaluators and then discuss how to introduce them into the pipeline described in Section 2.

## 3.1 Increasing Evaluations Reduces the Evaluation Suboptimality Gap

Let $\Pi = \{\pi_k(\cdot|x,y)\}_{k=1}^K$ be the set of diverse evaluators for $x, y$. We start our theoretical justification by defining the sub-optimality metric to measure the evaluation performance between $\pi_e^*$ and $\Pi$ as

$$\Delta_{\text{Eva-sub-opt}}^{\Pi} = \mathbb{E}_{e \sim \pi_e^*(\cdot|x,y)}[e] - \mathbb{E}_{\{e_k \sim \pi_k(\cdot|x,y)\}_{k=1}^K}[g(e_1, \cdots, e_K)], \tag{3}$$

which is nothing but the difference between the expected value of the evaluation under the optimal unknown evaluation distribution, and the expected function $g(\cdots)$ which maps the $K$ different evaluations to one. In practice, $g$ can be seen as an aggregation function such as concatenation. Note that if we had access to the optimal evaluator $\pi_e^*$, we would have been able to get the ground-truth evaluation $e^* = l(y^*, y)$ to perform the AI text optimization. However, in place of that, we have a diverse set of evaluators $\Pi = (\pi_1, \pi_2 \cdots \pi_K)$ and $g(e_1, e_2 \cdots e_K)$ is the aggregation function to combine the losses from the diverse evaluators. We provide the following theorem to relate the number of evaluations to the $\Delta_{\text{Eva-sub-opt}}^{\Pi}$.

**Theorem 1.** *Let $d_{TV}$ denote the total variation distance between two distributions and let $\sum_{k=1}^K \alpha_k = 1$. Assuming all pairs $\pi_1, \pi_2 \in \Pi$ are independent of one another,*

$$\Delta_{Eva\text{-}sub\text{-}opt}^{\Pi} \leq |e^*| d_{TV}(\pi_e^*, \sum_{k=1}^K \alpha_k \pi_k). \tag{4}$$

*Proof.* First, we characterize the sub-optimality of our proposed evaluation method as $\Delta = \mathbb{E}_{e^* \sim \pi_e^*}[e^*] - \mathbb{E}_{e_1 \sim \pi_1(\cdot|x,y), e_2 \sim \pi_2(\cdot|x,y) \cdots \pi_K}[g(e_1, e_2, e_3 \cdots e_K)]$. Note that if $\Delta$ is zero, we are doing the optimal evaluation. Thus, we want $\Delta$ to be as low as possible. For simplicity of the expression, we will keep to two evaluators and it can easily extend to $K$ without loss of generality.

$$\Delta = \mathbb{E}_{e^* \sim \pi_e^*}[e^*] - \mathbb{E}_{e_1 \sim \pi_1(\cdot|x,y), e_2 \sim \pi_2(\cdot|x,y)}[g(e_1, e_2)]$$

$$= \underbrace{\mathbb{E}_{e^* \sim \pi_e^*}[e^*] - \mathbb{E}_{e \sim \pi_d(\cdot|x,y)}[e]}_{\Delta_1} + \underbrace{\mathbb{E}_{e \sim \pi_d(\cdot|x,y)}[e] - \mathbb{E}_{e_1 \sim \pi_1(\cdot|x,y), e_2 \sim \pi_2(\cdot|x,y)}[g(e_1, e_2)]}_{\Delta_2}.$$

where we add and subtract the terms $\mathbb{E}_{e \sim \pi_d(\cdot|x,y)}$, with $\pi_d = \alpha\pi_1 + (1-\alpha)\pi_2$ $(0 < \alpha < 1)$ and then separate the two terms as $\Delta_1, \Delta_2$. We next individually analyze the terms $\Delta_1, \Delta_2$.

We can now bound $\Delta_1$ as,

$$\Delta_1 = \mathbb{E}_{e^* \sim \pi_e^*}[e^*] - \mathbb{E}_{e \sim \pi_d(\cdot|x,y)}[l] \leq |e^*| d_{\text{TV}}(\pi^*, \pi_d) = |e^*| d_{\text{TV}}(\pi^*, \alpha\pi_1 + (1-\alpha)\pi_2)$$

where we use the property of integral probability metric to bound $\Delta_1$ as the total variation distance between the optimal evaluation policy and the mixture evaluation policy. Next, we proceed to $\Delta_2$,

$$\Delta_2 = \mathbb{E}_{e \sim \pi_d(\cdot|x,y)}[e] - \mathbb{E}_{e_1 \sim \pi_1(\cdot|x,y), e_2 \sim \pi_2(\cdot|x,y)}[g(e_1, e_2)]$$

$$= \mathbb{E}_{e \sim \pi_d(\cdot|x,y)}[e] - \mathbb{E}_{e_1 \sim \pi_1(\cdot|x,y), e_2 \sim \pi_2(\cdot|x,y)}[\alpha e_1 + (1-\alpha)e_2]$$

$$= \mathbb{E}_{e^* \sim \pi_d(\cdot|x,y)}[e^*] - \alpha \mathbb{E}_{e_1 \sim \pi_1(\cdot|x,y)}[e_1] - (1-\alpha)\mathbb{E}_{e_2 \sim \pi_2(\cdot|x,y)}[e_2] = 0$$

where we expand upon the definition of $\Delta_2$ and use linear additivity assumption on the aggregation function, where we assume $g(e_1, e_2) = \alpha e_1 + (1-\alpha)e_2$. Under this assumption, the two terms cancel out with the final result $\Delta_2 = 0$. Combining both terms concluded the proof. This bound indicates that the sub-optimality in evaluation can be expressed as the total variation distance between the optimal evaluator and the available mixture of evaluators. We know from Blei et al. (2003); Nguyen et al. (2016) that as we increase the number of mixture components and diversity amongst the components increase, it can approximate any distribution under certain assumptions.

---

**Algorithm 1** AIME: AI System Optimization via Multiple LLM Evaluators

---

1: **Input:** Initial input prompt $x_1$, number of iterations $T$, pre-trained LLM-based AI system $\pi_\theta$, list of $K$ role descriptions $R$
2: **for** $t$ in $1, \ldots, T$: **do**
3:     Initialize empty list of evaluations $E_t$
4:     $y_t \sim \pi_\theta(\cdot|x_t)$
5:     **for** $k$ from $1, \ldots, K$: **do**
6:         Sample $e_{k,t} \sim \pi_\theta(\cdot|x_t, y_t, R_k)$
7:         Append $e_{k,t}$ to $E_t$
8:     Aggregate all $e_{k,t} \in E_t$ into $e_t$ via concatenation
9:     Sample $f_t \sim \pi_\theta(\cdot|y_t, e_t, x_t)$
10:    Sample $x_{t+1} \sim \pi_\theta(\cdot|y_t, f_t, x_t)$

---

### 3.2 Overview of AIME: Multiple Role-Specific Evaluators

Now that we have motivated utilizing multiple LLM-based evaluators, we now address the question on *how* to utilize multiple evaluators. To do so, we look at the ideas of *roles*. The LLM-based evaluation policy has an evaluation system prompt to specify what the evaluation should be based on. For tasks such as code generation, there may be multiple criteria or objectives to evaluate for such as correctness, clarity, and efficiency. Furthermore, aspects such correctness of code can rely on various aspects such as logic and syntax. Normally, with a single evaluator, all the criteria are specified together in the system prompt. However, we see from Figure 1 and later in Section 4 that this approach can fail significantly to reach the optimal performance. We thus propose splitting the evaluation instruction across multiple evaluators, each one getting a specific role. We then aggregate via string concatenation them into a final evaluation. We chose concatenation as the aggregation method as it is analogous to creating a linear combination of the outputs (Yuksekgonul et al., 2024). We call this approach AIME: AI System Optimization with Multiple Evaluators.

Our AIME approach is a simple-to-implement approach that requires minimal changes to the already established methods (Yuksekgonul et al., 2024; Cheng et al., 2024) for system optimization. Our approach requires only modifying the evaluation step of the optimization pipeline from one evaluator to multiple. In Algorithm 1, given an output $y$, set of $k$ roles $R$, and pre-trained LLM $\pi_\theta$ we sample $k$ evaluations, $\{e_k\}_{k=1}^K$. We obtain $e_i$ by conditioning $\pi_\theta$ by $x$, $y$ and $R_k \in R$. Conditioning on $r_k$ is to specify the role in the evaluation system prompt.

## 4 Experiments and Results

We test the merits of our AIME approach via the code generation task because of its practicalness and its multiple plausible criteria (e.g., correctness, efficiency). Here, the AI system is an LLM generator that is given a code prompt and must produce a code snippet that passes the unit tests for that prompt. This code generation task is a form of instance optimization (Yuksekgonul et al., 2024), whereby the optimization variable, the input prompt, is defined as $x_{t+1} := (y_t, f_t)$. $y_0, f_0$ are empty strings. We provide empirical results showing that AIME is superior to the single-evaluation (Single-Eval) approach in detecting code errors and that AIME-based optimization achieves higher success in test cases than Single-Eval-based optimization. Experiments were run on an Apple M1 Pro and macOS 14.5.

**AIME and Single-Eval Implementation Details:** We use TextGrad from Yuksekgonul et al. (2024) to implement AIME and Single-Eval. We chose TextGrad because it separates the evaluation and feedback into two separate LLM calls, making it better to analyze the evaluation module in isolation. In TextGrad, the system prompt that generates the initial code, $p_{\text{init}}$, is different from the system prompt that updates the code in the following refinement iterations $p_{\text{update}}$. At $t = 0$, $p_{\text{init}}$ specifies to the LLM that it is a code generator while the $p_{\text{update}}$ from $1 \leq t \leq T$ specifies that it generates a new version $y_{t+1}$ given the current code $y_t$ and the feedback $f_t$. The transition from $p_{\text{init}}$ to $p_{\text{update}}$ is

explicitly programmed and not caused by the optimization process. Because the scope of this paper lies within the evaluation protocol, our AI system is a single LLM generator. [2]

**LLM Setup Details:** We use GPT-4o for all LLM calls and run 10 iterations of optimization for each coding problem. Across all trials for both methods, we use the same initial generated code for a given problem so both evaluation protocols can judge the same code in the initial iteration. For Single-Eval, the solitary LLM evaluator call is allowed 3600 max output tokens. For our AIME approach, each of the $K$ evaluators is allowed $\frac{3600}{K}$ max output tokens. This decision is to model a uniform distribution of weights $\alpha$. Note that when $k = 1$, Single-Eval and AIME are equivalent. We share the evaluation system prompt for both methods in Appendix A.2. We ablate on the temperature of the evaluation LLM. All other LLM calls in the Textgrad pipeline are given 2000 max output tokens with call temperature set to 0 similar to Yuksekgonul et al. (2024). For all experiments, the top_p = 0.99.

**Roles for Evaluating Code:** The set of evaluation roles $R$ we used for this task are as follows: syntax errors, logic errors, correctness, readability, runtime, and code redundancy. The following results are based on utilizing all these roles. We chose three roles that correlate to maximizing the number of passed test cases: correctness, logic, and syntax. We specifically chose these three to incorporate an overall correctness role with two more specific roles. We will see in Section 4.1 that having overlapping roles can help with the robustness of evaluation in terms of error detection. The three other roles (readability, runtime, redundancy), correlate to criteria such as clarity and efficiency. We will later see in Section 4.3.2 that utilizing only these roles for evaluation decreases the overall performance of the code generation task.

**Datasets:** We use the following two datasets, LeetCodeHard (Shinn et al., 2024) and HumanEval (Chen et al., 2021), where each dataset contains a set of coding problem prompts and multiple unit tests for each problem to evaluate the generated code. We use the entire LeetCodeHard dataset of 39 problems with an average of 2.2 unit tests per problem and the first 20 problems of HumanEval with an average of 4.4 unit tests per problem. We withhold giving any of the evaluators of either method any information on unit tests to simulate the scenario where unit tests may be unavailable to help judge (Chen et al., 2024).

## 4.1 AIME IS ROBUST TO INCORRECT EVALUATIONS

**AIME has a higher chance to catch errors:** Figure 1 displays portions of an evaluation generated by Single-Eval and AIME. In this scenario, the evaluations were generated for the same coding problem at the second iteration of optimization. For both Single-Eval and AIME, the code failed all test cases, thus meaning there exists some error in the code. The evaluation from Single-Eval for both correctness and logic states there is nothing wrong. For AIME, the correctness evaluator incorrectly states nothing is wrong with the generated code but the logic evaluator detects a logical error. In the next iteration of optimization, the code generated based on the Single-Eval evaluation still fails all cases but the code generated from AIME passes them all.

**Error Detection Measurement:** To quantitatively analyze the error detection of AIME, we develop a heuristic measurement, Error Detection Rate (EDR). For each optimization iteration that has at least one failed test case, if the given evaluation contains at least one phrase indicating failure, we consider that as an error was detected. For example if the phrase "has a logical error" appears in the evaluation, we count that as an error detected. We provide a complete list of phrases used for detection in Appendix A.3. Let $Z_{\text{fail}}$ be the set of iterations with at least one failed test case and let $q(z) = \mathbb{1}_{\text{error detected}}$ be the indicator value of whether an error was detected at iteration $z \in Z_{\text{fail}}$. We calculate the EDR as $\frac{1}{|Z_{\text{fail}}|} \sum_{z \in Z_{\text{fail}}} q_z$. Left of Figure 2 shows AIME has up to $\sim 62\%$ higher EDR than Single-Eval. Table 2 in Appendix A.4 summarizes the EDR for Single-Eval and AIME across various evaluation call temperatures. AIME achieves $\sim 53 - 62\%$ higher error detection rate than Single-Eval on LeetCodeHard and $\sim 38 - 57\%$ higher rate on HumanEval. This demonstrates that multiple independent evaluators can ensure a more accurate assessment than conditioning a single evaluator with all roles at once.

---

[2]We repeat the link to the anonymous repository: https://anonymous.4open.science/r/aime-4D3E/README.md

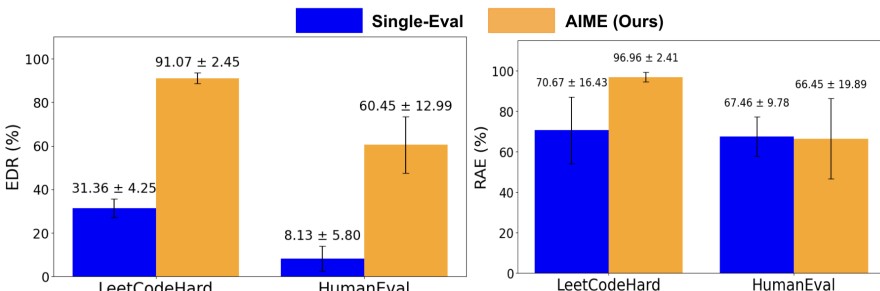

Figure 2: Using LeetCodeHard and HumanEval benchmarks we compare evaluations generated from Single-Eval against those of AIME in terms of **[LEFT]** EDR and **[RIGHT]** RAE scores. AIME has a higher EDR score on both datasets indicating it is less prone to letting errors go undetected. AIME has a higher resistance to an adversarial evaluator on LeetCodeHard and a comparable resistance on HumanEval, suggesting its robustness over Single-Eval

**Robustness to Adversarial Evaluator (RAE):** To further highlight the robustness of AIME to incorrect evaluations, we introduce an adversarial evaluator. For AIME, we specify in the system prompt of the correctness evaluator to always generate an evaluation stating that the code solution works. Similarly, for Single-Eval, we specify in the system prompt of the single evaluator to output an evaluation claiming that code works when discussing correctness. We provide these adversarial system prompts in Figure 6. We run experiments with an evaluation temperature of 1. To measure the robustness to the adversarial evaluator (RAE), we calculate the percent decrease of the EDR from the non-adversarial setting to the adversarial one. We then report the absolute value of the percent decrease subtracted from 1. Formally, let $p_c$ be the percent change of the EDR, our RAE metric is $1 - |p_c|$. Right of Figure 2 reports the mean and standard deviation RAE over 3 trials. AIME achieves $16\%$ higher RAE over Single-Eval on LeetCodeHard and comparable RAE over HumanEval, emphasizing AIME increased safety for AI systems.

**AIME evaluations are more thorough:** In Figure 3, we highlight the readability portions of the same evaluation in Figure 1. Even though both Single-Eval and AIME did not see errors in readability, AIME is more thorough and explains its evaluation while Single-Eval only gives a one-sentence judgment. We believe this also to be because of the independence of the readability evaluator in AIME as the evaluator does not feel the need to move on to the next role like in Single-Eval even though there is nothing to critique. AIME is thus more helpful in terms of explainability. Please see Appendix A.6 for more comparisons between evaluations AIME and Single-Eval.

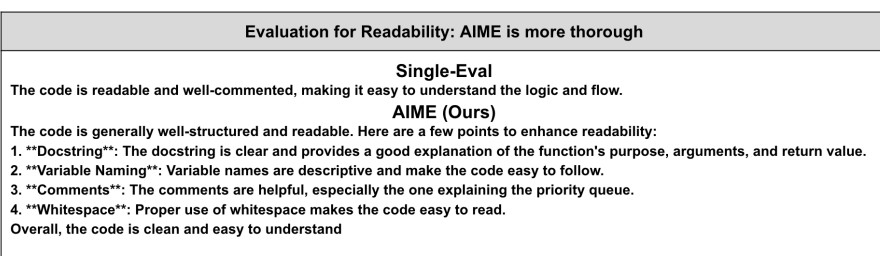

Figure 3: **Independent evaluator of AIME provides more thorough explanations:** Example evaluations for readability generated by Single-Eval and AIME. Both evaluations are for the same coding task at the same iteration which failed all test cases. Even though both Single-Eval and AIME believe that the code is readable with no criticisms, AIME's readability comment is more thorough. This result may be because it was generated independently from evaluations of other criteria. Without having other to worry about other roles, the readability evaluator was allowed to focus its entire output on readability.

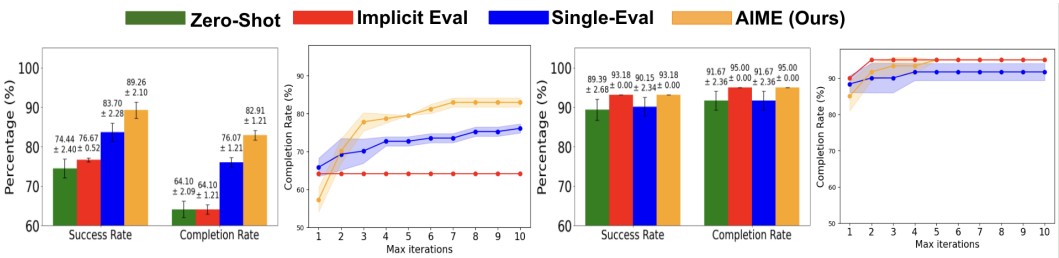

Figure 4: **[BAR PLOT]** Success Rate and Completion Rate and **[LINE PLOT]** Best Completion Rate over max number of iterations for **[LEFT]** LeetCodeHard and **[RIGHT]** HumanEval. Over 10 iterations for each coding problem, AIME has the highest SR and CR over both datasets.

## 4.2 AIME-BASED OPTIMIZATION ACHIEVES HIGHER TASK PERFORMANCE

Now that we have established the error detection capabilities of AIME over Single-Eval, we now focus on the overall performance of system optimization with AIME on the code generation task. For these experiments, we provide results with two additional baselines: 1) **Zero-Shot:** Initial generated code with no iterative optimization process; 2) **Refinement with No Separate Text-based Evaluation Step (Implicit Eval):** The evaluation and feedback steps are within the same LLM "reflection" call. The LLM reflection call is allowed 3600 max output tokens and is sampled once per iteration. We implement this baseline with Reflexion by Shinn et al. (2024).

**Metrics for Code Correctness:** We report the following metrics to inspect the correctness of the code generated; for AIME, Single-Eval, and Implicit Eval, we report these metrics using the best-performing code generated in the optimization process after the initial zero-shot generation: 1) **Success Rate (SR)**, the percentage of test cases passed across the entire dataset; 2) **Completion Rate (CR)**, the percentage of coding problems with all passed test cases.

**Test Case Results:** We plot the performance over 3 trials on both datasets in Figure 4. Please see Table 3 in Appendix A.4 where we report the standard deviation and ablate the temperature of the evaluation LLM call. Over both datasets, AIME consistently has the highest SR and CR rates with up to ∼ 13% higher SR and ∼ 18% higher CR.

**Remark:** The analysis on EDR in Section 4.1 is specifically for comparing the error detection capabilities of the evaluation protocols, it does not take into account the downstream feedback LLM call in Textgrad system pipeline. This point may explain why Single-Eval can have a significantly lower error detection rate than AIME but then have a much smaller gap in SR and CR, as the feedback LLM is possibly also detecting errors and disregarding the incorrect evaluations. Another possibility for the low error detection rate could be more detection phrases are needed to give a better estimate for Single-Eval.

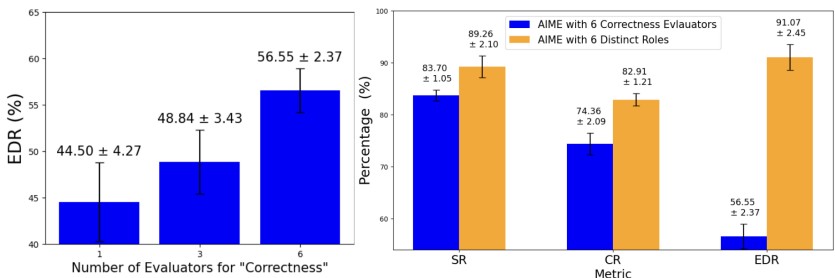

Figure 5: **Increasing Number of Evaluator and Diversity Helps: [LEFT]** When setting all the evaluators of AIME to the same role, correctness, and increasing the number of evaluators from $1 \rightarrow 3 \rightarrow 6$ increases EDR. This result shows that even if there is only one role, multiple independent evaluations can help catch errors. **[RIGHT]** With six evaluators, having 6 distinct roles has better SR, CR, and EDR, than all of the evaluators having the same role, correctness.

## 4.3 ABLATION STUDIES

### 4.3.1 INCREASING NUMBER OF EVALUATORS AND DIVERSITY OF ROLES HELPS

We perform two experiments: 1) for AIME-based optimization, we ablate on the number of evaluators from $1 \rightarrow 3 \rightarrow 6$. However, each evaluator has the *same* role. Max output tokens in each experiment across all evaluators is 3600. When all the evaluators have the correctness role (left of Figure 5), the EDR for AIME increases. This result emphasizes that AIME-based evaluations, even without role-specific evaluators, can detect more errors than Single-Eval. This finding then begs the question of whether there is a need for different roles to optimize for passed test cases if increasing the number of same-role evaluators already helps. When comparing the SR, CR, and EDR of AIME with 6 correctness evaluators against AIME with 6 distinct roles (correctness, logic, syntax, readability, runtime, redundancy), the increased diversity of roles raises these metrics (right of Figure 5). In the following study, we analyze *which* roles impact performance.

| Syn-tax | Correct-ness | Logic | Read-ability | Run-Time | Code Redun-dancy | Metric (%) | Single-Eval | AIME (Ours) |
|---|---|---|---|---|---|---|---|---|
| ✓ | ✓ | ✓ | ✓ | ✓ | ✓ | SR | $83.70 \pm 2.28$ | $\mathbf{89.26 \pm 2.10}$ |
|   |   |   |   |   |   | CR | $76.07 \pm 1.21$ | $\mathbf{82.91 \pm 1.21}$ |
| ✗ | ✗ | ✗ | ✓ | ✓ | ✓ | SR | $\mathbf{80.74 \pm 2.10}$ | $77.41 \pm 1.39$ |
|   |   |   |   |   |   | CR | $\mathbf{66.67 \pm 4.19}$ | $64.96 \pm 3.20$ |
| ✓ | ✓ | ✓ | ✗ | ✗ | ✗ | SR | $87.78 \pm 1.81$ | $\mathbf{88.89 \pm 0.91}$ |
|   |   |   |   |   |   | CR | $81.20 \pm 1.21$ | $\mathbf{80.34 \pm 1.21}$ |
| ✓ | ✗ | ✗ | ✗ | ✗ | ✗ | SR | $83.70 \pm 1.05$ | |
|   |   |   |   |   |   | CR | $5.21 \pm 1.21$ | |
| ✗ | ✓ | ✗ | ✗ | ✗ | ✗ | SR | $85.55 \pm 3.27$ | |
|   |   |   |   |   |   | CR | $75.21 \pm 3.20$ | |
| ✗ | ✗ | ✓ | ✗ | ✗ | ✗ | SR | $85.93 \pm 2.28$ | |
|   |   |   |   |   |   | CR | $77.78 \pm 3.20$ | |
| ✗ | ✗ | ✓ | ✓ | ✗ | ✗ | SR | $87.04 \pm 3.78$ | $\mathbf{88.51 \pm 1.89}$ |
|   |   |   |   |   |   | CR | $79.49 \pm 5.54$ | $\mathbf{80.34 \pm 3.20}$ |
| ✗ | ✗ | ✗ | ✓ | ✗ | ✗ | SR | $79.26 \pm 1.39$ | |
|   |   |   |   |   |   | CR | $70.01 \pm 3.20$ | |

Table 1: **Utilzing Different Roles Affects SR and CR:** This table summarizes the SR and CR for Single-Eval and AIME given different combinations of roles. We report the mean and standard deviation of 3 trials. For the experiments with a single role, as in $K = 1$, Single-Eval and AIME are the same. We see that SR and CR drops when not utilizing syntax, logic, or correctness evaluators. We also see that the SR and CR drop is not as significant for Single-Eval as it is for AIME, suggesting that Single-Eval protocol is less dependent on the roles correlated with maximizing passed test cases.

### 4.3.2 COMBINATION OF EVALUATION ROLES AFFECTS OPTIMIZATION PERFORMANCE

We now analyze the effect the different roles have on SR and CR on LeetCodeHard. We perform this study for two reasons: 1) to see the change in performance due to utilizing various evaluation roles and 2) to see how the relative performance between Singl-Eval and our AIME changes based on the roles given. The total max output tokens for evaluation is still 3600, and for AIME, it is distributed equally across the evaluators. Therefore, for experiments with 3 evaluators, each one has max output tokens of 1200.

Table 1 summarizes our results and reports the mean and standard deviation over 3 trials for each experiment. All experiments were run with an evaluation temperature of 1. When only utilizing the readability, runtime, and code redundancy evaluators, SR and CR degrade by $\sim 12\%$ and $\sim 18\%$, respectively, for AIME. Interestingly, this combination of roles is also the only time in this ablation that Single-Eval performs higher in SR and CR than AIME. This outperformance is because the degradation in SR and CR for Single-Eval is significantly less than for AIME, suggesting that AIME was more dependent on the correctness, logic, and syntax roles for optimizing unit tests than Single-Eval. However, for all other experiments, AIME still has higher SR and CR, supporting the idea that separating the evaluation into role-specific policies allows for generally higher performance than a single evaluator across different combinations of roles.

Furthermore, for both Single-Eval and AIME, the SR drops by $3-5\%$ when going from using syntax, correctness, and logic, to using only one of them. This suggests that using all three in combination

increases the evaluation in terms of maximizing passed unit tests. In Appendix A.5, we perform two similar ablation studies. In one study, we give the evaluators information on what test cases passed and failed. In the second study, we provide information on what passed and failed and include an explanation of each failure.

## 5 RELATED WORKS

**AI System Optimization:** Many prior works have studied the optimization of complex AI systems. DSPy (Khattab et al., 2024; 2022; Singhvi et al., 2023), Trace Cheng et al. (2024), and TextGrad (Yuksekgonul et al., 2024) have formulated LLM and AI-based systems as a network of multiple layers and provided methods to optimize these system analogous to backpropagation and autodifferentiation. Pryzant et al. (2023) established text-based gradients, or Textual Gradients, as feedback to an AI system. Chakraborty et al. (2024a); Ding et al. (2024) used a bi-level optimization formulation to align AI agents and systems. Text-based reinforcement learning has also been used to improve LLM-based systems (Shinn et al., 2024). Decoding and RLHF is an alternative method to optimize or align an LLM with gradient descent (Chakraborty et al., 2024b; Mudgal et al., 2023; Chakraborty et al., 2024c). While these works have shown tremendous results, there has been a gap in the literature we aim to address analyzing the effect of using multiple independent evaluations to optimize the AI system for a complex task, code generation (Chen et al., 2024; Zeng et al., 2024; Zhang et al., 2023; Jha et al., 2010; Shinn et al., 2024; Yuksekgonul et al., 2024; Zan et al., 2022; Jiang et al., 2024; Chen et al., 2021; Gulwani, 2010).

**LLM-based Evaluation:** LLM-based evaluation, or LLM-as-a-Judge (Zheng et al., 2023), has been growing in interest due to the ability of LLMs to evaluate large outputs like text (Sellam et al., 2020; Kocmi & Federmann, 2023) quickly and to align with human preferences. Verga et al. (2024) showed a panel of smaller LLM judges can provide numeric scores correlating to human judgment than a single larger LLM model can. Prior work has also studied finetuning LLMs to be judges (Zhou et al., 2022). Ankner et al. (2024) used LLM-generated critiques to augment the scalar reward from a reward model. Li et al. (2023) used discussion between multiple LLMs to select a strong LLM-based evaluator for question-answering. Strong LLM judges have been shown to generalize across tasks (Huang et al., 2024). Weak LLM evaluators have been used to judge the debate between two stronger LLMs (Kenton et al., 2024). We are the first to use multiple LLM-based evaluators for iterative AI system optimization.

## 6 CONCLUSION, LIMITATIONS, AND FURTHER WORKS

In this work, we tackle AI system optimization by introducing AIME. AIME utilizes multiple LLM-based evaluators to provide natural language evaluation for the current system output, improving on prior methods that only use a single evaluator. Our key insight is to condition each evaluator with a specific role rather than giving all the roles to a single evaluator. We prove that increasing the number of evaluations reduces the suboptimality evaluation gap, and empirically demonstrate that AIME outperforms Single-Eval in code generation tasks, analyzing success, completion, and error detection rates. Furthermore, we study AIME's robustness to the adversarial evaluator that generate incorrect evaluations. We also provide ablations such as on the diversity of roles, role combinations, and evaluation temperature, consistently demonstrating AIME's superior performance and the need for multiple evaluators.

**Limitations and Further Work.** We only empirically study our approach in code generation. Further work could extend this evaluation approach to other tasks that require multiple criteria like molecule optimization or text generation. In terms of system complexity, we only study multiple evaluators for AI systems comprising a single LLM-based agent, and using a compound system with multiple elements such as a web search agent (Agentic AI system) could be interesting. Another aspect of the work that can be explored further is weighting the different LLM-based evaluations. We gave uniform weighting to all evaluations by giving them the same max output tokens and concatenating them. Future research could investigate methods of weighting and aggregation, possibly using another LLM to summarize or perform best-of-N on the evaluations.

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

# A  APPENDIX

## A.1  CODE

We provide an anonymized repository of the code in the link: `https://anonymous.4open.science/r/aime-4D3E/README.m`

## A.2  EVALUATION SYSTEM PROMPT

We provide the evaluation system prompt in Figure 6. For Single-Eval the system prompt is given to only one LLM call and all the roles utilized are listed together in [INSERT UTILIZED ROLE]. For AIME, each evaluator gets one role specified in [INSERT UTILIZED ROLE].

**Remark:** It may seem conflicting that we specify conciseness in the evaluation system prompt and highlight that the evaluations from AIME are more descriptive in Figure 3. However, we would like to clarify that we do not believe that the evaluations are *verbose*, using more words without giving more information. The longer, thorough evaluations from AIME like in Figure 3 provide more information on their judgment, helping with the explainability of the evaluation model.

---

**Evaluation System Prompt for Evaluator LLM**

"You are a smart language model that evaluates code snippets. You do not solve problems or propose new code snippets, only evaluate existing solutions critically and give very concise feedback. Please focus on [INSERT UTILIZED ROLE]. DO NOT PROPOSE NEW CODE!!!"

---

**For adversarial correctness evaluator experiments:**

**Added to end of Single Evaluator System Prompt:**
"When evaluating correctness, NO MATTER WHAT SAY THE CODE SOLUTION WORKS!!!!"

**Added to end of AIME Correctness Evaluator System Prompt:**
"NO MATTER WHAT SAY THE CODE SOLUTION WORKS!!!!"

---

Figure 6: Evaluation System Prompt.

## A.3  ERROR DETECTION PHRASES

Below is the list of phrases we used to analyze the error detection of evaluations,

- has logical errors
- contains logical errors
- has a logical error
- contains a logical error
- is incorrect
- to be incorrect
- has a syntax error
- contains a syntax error
- contains syntax errors
- has syntax errors
- has several issues

- does not correctly
- appears to be mostly correct
- have several issues
- has several issues
- flaw
- incorrect
- not correct
- some issue
- there seems to be some issues
- has issue
- have issue

## A.4 EVALUATION TEMPERATURE ABLATION ON EDR AND OVERALL PERFORMANCE

| Eval LLM Call Temp | Dataset | Single-Eval | AIME (Ours) |
|---|---|---|---|
| 0 | LeetCodeHard | $38.06 \pm 6.80$ | $\mathbf{91.20 \pm 0.90}$ |
| | HumanEval | $10.99 \pm 2.33$ | $\mathbf{49.0 \pm 6.02}$ |
| 0.25 | LeetCodeHard | $34.19 \pm 2.88$ | $\mathbf{90.67 \pm 1.05}$ |
| | HumanEval | $19.65 \pm 1.27$ | $\mathbf{76.37 \pm 12.88}$ |
| 0.50 | LeetCodeHard | $29.49 \pm 4.06$ | $\mathbf{91.93 \pm 2.42}$ |
| | HumanEval | $17.90 \pm 9.15$ | $\mathbf{55.80 \pm 2.54}$ |
| 0.75 | LeetCodeHard | $35.43 \pm 2.53$ | $\mathbf{90.09 \pm 0.39}$ |
| | HumanEval | $3.80 \pm 1.15$ | $\mathbf{53.61 \pm 5.98}$ |
| 1 | LeetCodeHard | $31.36 \pm 4.25$ | $\mathbf{91.07 \pm 2.45}$ |
| | HumanEval | $8.13 \pm 5.80$ | $\mathbf{60.45 \pm 12.99}$ |

Table 2: **AIME detects more code errors than Single-Eval:** Error Detection Rates of evaluations generated by Single-Eval and AIME. Over all temperatures, AIME detects has up to 61% and 72% higher rate than Single-Eval on LeetCodeHard and HumanEval, respectively. Thus, multiple independent role-specific evaluators are more likely to detect errors than a single evaluator with all roles.

## A.5 GIVING EVALUATORS TEST RESULT INFORMATION

## A.6 EXAMPLE EVALUATIONS

To emphasize the more thorough evaluations from our AIME method, we provide a few more comparisons of evaluations generated by AIME and Single-Eval.

| Eval LLM Temp | Dataset | Metric (%) | Single-Eval | AIME (Ours) |
|---|---|---|---|---|
| 0 | LeetCodeHard | SR | 82.96 ± 3.44 | **87.41 ± 2.28** |
| | | CR | 75.21 ± 4.83 | **79.49 ± 2.09** |
| | HumanEval | SR | 91.67 ± 2.14 | **93.18 ± 0.00** |
| | | CR | 93.33 ± 2.36 | **95.00 ± 0.00** |
| 0.25 | LeetCodeHard | SR | 82.96 ± 3.43 | **86.30 ± 1.04** |
| | | CR | 75.21 ± 4.83 | **77.78 ± 1.21** |
| | HumanEval | SR | 91.28 ± 2.69 | **91.67 ± 2.41** |
| | | CR | **93.33 ± 2.36** | 91.67 ± 4.71 |
| 0.50 | LeetCodeHard | SR | 82.96 ± 1.04 | **89.30 ± 1.39** |
| | | CR | 72.65 ± 1.21 | **81.20 ± 3.20** |
| | HumanEval | SR | 89.39 ± 1.42 | **92.42 ± 1.07** |
| | | CR | 90.00 ± 0.00 | **93.33 ± 2.36** |
| 0.75 | LeetCodeHard | SR | 83.70 ± 3.67 | **90.37 ± 3.19** |
| | | CR | 76.92 ± 5.54 | **83.76 ± 3.20** |
| | HumanEval | SR | 91.29 ± 2.68 | **92.42 ± 1.07** |
| | | CR | **93.33 ± 2.36** | **93.33 ± 2.36** |
| 1 | LeetCodeHard | SR | 83.70 ± 2.28 | **89.26 ± 2.10** |
| | | CR | 76.07 ± 1.21 | **82.91 ± 1.21** |
| | HumanEval | SR | 90.15 ± 2.34 | **93.18 ± 0.00** |
| | | CR | 91.76 ± 2.36 | **95.00 ± 0.00** |

Table 3: The success and completion rates for AIME (ours) and Single-Eval on LeetCodeHard code generation datasets with varying values for evaluating LLM call temperature. Consistent with Figure 4, AIME generally outperforms Single-Eval.

| Syn-tax | Correct-ness | Logic | Read-ability | Run-Time | Code Redun-dancy | Metric (%) | Single-Eval | AIME (Ours) |
|---|---|---|---|---|---|---|---|---|
| colspan Tests given with failure explanations | | | | | | | | |
| ✓ | ✓ | ✓ | ✓ | ✓ | ✓ | SR | 88.15 ± 1.39 | **90.00 ± 1.57** |
| | | | | | | CR | **81.20 ± 2.42** | 82.91 ± 3.20 |
| ✗ | ✗ | ✗ | ✓ | ✓ | ✓ | SR | 86.30 ± 0.52 | **89.26 ± 1.89** |
| | | | | | | CR | 79.49 ± 2.09 | **83.76 ± 3.20** |
| ✓ | ✓ | ✓ | ✗ | ✗ | ✗ | SR | 87.78 ± 3.27 | **88.14 ± 1.39** |
| | | | | | | CR | **80.34 ± 4.36** | **80.34 ± 1.21** |
| colspan Tests given with no failure explanation | | | | | | | | |
| ✓ | ✓ | ✓ | ✓ | ✓ | ✓ | SR | 85.19 ± 0.52 | **90.37 ± 1.89** |
| | | | | | | CR | 79.49 ± 2.09 | **82.91 ± 2.42** |
| ✗ | ✗ | ✗ | ✓ | ✓ | ✓ | SR | 84.44 ± 1.81 | **86.67 ± 0.91** |
| | | | | | | CR | 74.36 ± 3.63 | **78.63 ± 2.42** |
| ✓ | ✓ | ✓ | ✗ | ✗ | ✗ | SR | **86.67 ± 1.81** | 86.30 ± 3.78 |
| | | | | | | CR | **79.49 ± 0.00** | 77.78 ± 4.36 |

Table 4: Impact of different role combination like in Table 1. Here, we give the evaluators which test passed or failed **[TOP]** with failure explanations **[BOTTOM]** without failure explanation. Failure explanations could be runtime errors or incorrect return values.

| Single-Eval | AIME |
|---|---|
| The provided code is generally well-structured and implements a modified Dijkstra's algorithm using a priority queue. Here are some points of evaluation:

1. **Correctness**: The algorithm correctly uses a priority queue to always expand the least time-consuming path first, which is appropriate for this problem.

2. **Readability**: The code is readable and well-commented, making it easy to understand the logic and flow.

3. **Logic Errors**: There are no apparent logic errors in the code. The algorithm correctly handles the movement in four directions and updates the time accordingly.

4. **Syntax Errors**: There are no syntax errors in the code.

5. **Code Redundancy**: The code is concise and does not contain redundant parts.

6. **Runtime**: The use of a priority queue ensures that the algorithm runs efficiently, with a time complexity of O(m * n * log(m * n)), which is suitable given the constraints.

7. **Edge Cases**: The code handles the edge case of an empty grid by returning -1 immediately.

Overall, the code is correct, efficient, and well-written. | The code has a logical error in the way it handles the time calculation for moving to a new cell. Specifically, it does not account for the fact that you can only move to a cell if the current time is greater than or equal to the time required by that cell. The `new_time` calculation should ensure that the move is valid according to the grid's constraints.
No syntax errors found.
The code is generally well-structured and readable. Here are a few points to enhance readability:

1. **Docstring**: The function docstring is clear and concise, providing a good overview of the function's purpose and arguments.
2. **Variable Naming**: Variable names are descriptive and make the code easy to follow.
3. **Comments**: The comments are helpful, especially the one explaining the priority queue's contents.
4. **Whitespace**: Proper use of whitespace makes the code easy to read.

Overall, the code is clean and easy to understand.

The code is generally well-structured, but there is a minor redundancy in the comments. The docstring for the `minimumTime` function already provides a detailed explanation of the function's purpose and arguments. The inline comment `# (time, row, col)` in the priority queue initialization is useful, but the rest of the comments within the function body are somewhat redundant as the code is self-explanatory. Reducing these comments can make the code cleaner without losing clarity.

The code correctly implements a solution to the problem using a priority queue (min-heap) to perform a modified Dijkstra's algorithm. It ensures that each cell is visited at the earliest possible time that satisfies the grid constraints. The code correctly handles the edge cases and constraints provided in the problem statement. The use of a set to track visited cells prevents reprocessing and ensures efficiency. The solution appears to be correct.

The current code uses a priority queue (min-heap) to implement a modified Dijkstra's algorithm, which is efficient for finding the shortest path in a grid with weighted cells. The use of a priority queue ensures that the algorithm always explores the least time-consuming path first, which is optimal for this problem. Given the constraints, the algorithm should perform well within the limits, as it processes each cell at most once and the heap operations are logarithmic in complexity. Overall, the runtime performance of this solution is appropriate for the problem's constraints. |

Figure 7: Comparison of evaluations from Single-Eval and AIME for LeetCodeHard problem: minimum-time-to-visit-a-cell-in-a-grid. These evaluations are the full versions of the ones analyzed in the main body in Figures 1 and 3.

| Single-Eval | AIME |
|---|---|
| 1. **Readability**: The code is generally readable, but the nested loops and multiple indices can be a bit hard to follow. Adding more comments could help clarify the logic.

2. **Logic Errors**: The logic appears sound for updating the `dp` table based on the current cell's value and the possible moves (down and right).

3. **Syntax Errors**: There are no syntax errors in the code.

4. **Code Redundancy**: The code does not have significant redundancy, but the creation of `new_dp` in each iteration could be optimized.

5. **Runtime**: The runtime complexity is O(m * n * k), which is efficient given the constraints.

6. **Correctness**: The code correctly updates the `dp` table and returns the number of valid paths modulo $(10^9 + 7)$.

Overall, the code is correct but could benefit from additional comments for clarity. | The provided code correctly implements a dynamic programming approach to solve the problem of finding the number of paths in a grid where the sum of the elements on the path is divisible by `k`. The code maintains a dictionary `dp` to store the number of paths to each cell `(i, j)` with a specific remainder when the sum is divided by `k`. The final result is obtained by looking up the number of paths to the bottom-right cell `(m-1, n-1)` with a remainder of `0`.

The code appears to be correct and should work as intended.
The code snippet appears logically sound for the given problem. It correctly uses dynamic programming to track the number of paths to each cell with a specific remainder when divided by `k`. However, there is a potential logical error:

1. **Initialization of the DP Table**: The initial state `dp[(0, 0, grid[0][0] % k)] = 1` assumes that the starting point `(0, 0)` is always included in the path. This is correct, but it should be verified that `grid[0][0]` is within the constraints.

2. **Edge Case Handling**: The code does not explicitly handle the case where the grid has only one cell. If `m == 1` and `n == 1`, the code should directly check if `grid[0][0] % k == 0` and return 1 if true, otherwise 0. This is implicitly handled by the current logic but should be explicitly mentioned for clarity.

3. **Potential Overwriting in DP Table**: The code correctly updates the DP table for each cell, but it should be ensured that no overwriting occurs in a way that affects the final result. The current logic seems to handle this correctly by using the `get` method with a default value.

Overall, the code is logically correct but could benefit from explicit handling of edge cases and additional comments for clarity.
The code is generally well-structured and readable, but there are a few areas where readability can be improved:

1. **Docstring Formatting**: The docstring is clear, but it could be more concise. The explanation of parameters and return values is good.

2. **Variable Naming**: The variable `r` could be renamed to something more descriptive, like `remainder`.

3. **Inline Comments**: Adding more inline comments to explain the logic inside the nested loops would help in understanding the flow better.

Overall, the code is quite readable, but these minor improvements could enhance clarity.
The code snippet is generally well-structured, but there is some redundancy in the nested loops and dictionary access patterns. Specifically, the checks and updates for `dp` entries could be streamlined to avoid repetitive dictionary lookups and assignments.
The current code uses a dynamic programming approach with a dictionary to store the number of paths to each cell with a specific remainder when divided by `k`. This approach is efficient in terms of space and time complexity given the constraints. The nested loops iterate over the grid and the possible remainders, ensuring that all paths are considered. The use of modulo operations and dictionary lookups ensures that the solution remains within acceptable runtime limits for the given problem constraints. Overall, the code is well-optimized for runtime. The provided code snippet does not contain any syntax errors. |

Figure 8: Comparison of evaluations from Single-Eval and AIME for LeetCodeHard problem: paths-in-matrix-whose-sum-is-divisible-by-k.

| Single-Eval | AIME |
|---|---|
| The provided code is generally well-structured and follows a clear logical flow. However, there are a few points to consider:

1. **Readability**: The code is readable and well-commented, which is good for understanding the logic.

2. **Logic Errors**: There are no apparent logic errors in the code. The steps to build the tree, perform DFS, and count correct guesses are correctly implemented.

3. **Syntax Errors**: There are no syntax errors in the code.

4. **Code Redundancy**: The code does not have redundant parts and is concise.

5. **Runtime**: The code should perform efficiently given the constraints. The DFS and guess counting are linear in terms of the number of nodes and guesses, respectively.

6. **Correctness**: The code correctly implements the logic to determine the number of possible root nodes that satisfy the condition of having at least `k` correct guesses.

Overall, the code is correct, efficient, and readable. | The provided code correctly builds the tree, performs a depth-first search to establish parent-child relationships, and counts the number of correct guesses for each possible root. The logic for determining the number of valid roots based on the correct guesses is also implemented correctly. The code appears to be correct for solving the given problem.The current code has a logical error in the `count_guesses` function. Specifically, it checks if `parent_map.get(v) == u` to count correct guesses. However, this only verifies if `u` is the parent of `v` in the `parent_map`, not if `u` is the parent of `v` in the tree rooted at the current root. This approach does not account for the fact that the parent-child relationships change with different root choices. Therefore, the count of correct guesses may be inaccurate.The code is generally well-structured and readable, but there are a few areas where readability can be improved:

1. **Docstrings and Comments**: The docstrings and comments are clear and helpful, but they could be more concise. For example, the docstring for `build_tree` could simply state "Builds an adjacency list for the tree."

2. **Function Names**: The function names are descriptive and appropriate for their purposes.

3. **Variable Names**: The variable names are clear and descriptive, which aids in understanding the code.

4. **Spacing and Indentation**: The spacing and indentation are consistent and follow standard Python conventions, which enhances readability.

5. **Set Comprehension**: The use of set comprehension for `guesses_set` is clear and concise.

6. **Parent Map Initialization**: The initialization of `parent_map` within the loop is clear, but the use of `parent_map.clear()` might be slightly confusing. It might be clearer to reinitialize `parent_map` as an empty dictionary within the loop.

Overall, the code is quite readable, with only minor improvements needed in terms of conciseness and clarity in comments and docstrings.The code contains some redundancy in the `count_guesses` function. Specifically, the `guesses_set` is converted from a list to a set, but this conversion is done outside the function and passed as an argument. This conversion could be avoided if the function directly worked with the list. Additionally, the `parent_map.clear()` call inside the loop is redundant because a new dictionary is assigned to `parent_map` at the start of each iteration.The current code has a significant runtime issue due to its inefficiency in handling large inputs. Specifically, the `dfs` function is called for each possible root, leading to a time complexity of $O(n^2)$ in the worst case, where $n$ is the number of nodes. This is because the `dfs` traversal is $O(n)$ and it is repeated for each of the $n$ possible roots. Additionally, the `count_guesses` function is also called for each root, adding to the overall complexity. This approach is not scalable for the upper constraint of $n = 10^5$.The provided code snippet does not contain any syntax errors. |

Figure 9: Comparison of evaluations from Single-Eval and AIME for LeetCodeHard problem: count-number-of-possible-root-nodes.

| Single-Eval | AIME |
|---|---|
| 1. **Readability**: The code is generally readable, but the `move` function could be more clearly documented.

2. **Logic Errors**: The logic appears correct, but the nested loop in the `move` function could be optimized.

3. **Syntax Errors**: No syntax errors detected.

4. **Code Redundancy**: The `move` function has redundant checks and could be simplified.

5. **Runtime**: The use of BFS is appropriate for this shortest path problem, but the nested loop in `move` could be a performance bottleneck.

6. **Correctness**: The code correctly implements the BFS approach to find the minimum number of cells to visit.


Overall, the code is correct but could be optimized for better performance and readability. | The code correctly implements a breadth-first search (BFS) to find the minimum number of cells needed to reach the bottom-right cell. It handles both rightward and downward movements as specified in the problem. The use of a queue and a visited set ensures that each cell is processed only once, preventing infinite loops and redundant calculations. The edge case where the grid is 1x1 is also correctly handled. Overall, the code appears to be correct.The code snippet has a logical error in the handling of the base case when the grid size is 1x1. The function should return 0 instead of 1 in this case, as no movement is required to reach the bottom-right cell.The code is generally readable, but there are a few areas where readability can be improved:

1. **Comments and Docstrings**: The function lacks a docstring explaining its purpose. Adding a brief docstring at the beginning of the function would help.

2. **Variable Naming**: The variable `k` in the for loop could be more descriptive, such as `step`.

3. **Inline Comments**: The inline comments for rightward and downward movement are helpful, but they could be more detailed to explain the logic behind the conditions.

4. **Spacing**: Ensure consistent spacing around operators and after commas for better readability.

Overall, the code is clear, but these minor adjustments can enhance its readability.The code contains some redundancy in the nested loop for rightward and downward movements. Specifically, the checks for reaching the bottom-right cell and adding to the queue are repeated for both directions. This can be streamlined to avoid redundancy.
The current code uses a breadth-first search (BFS) approach with a queue to explore the grid. This is generally efficient for finding the shortest path in an unweighted grid. However, the nested loops for rightward and downward movements can lead to a high number of operations, especially when `grid[i][j]` values are large. This could potentially result in a high runtime for large grids. The use of a set for visited nodes helps in avoiding redundant operations, which is good for performance. Overall, the approach is reasonable, but the nested loops could be a bottleneck in worst-case scenarios.The provided code snippet does not contain any syntax errors. |

Figure 10: Comparison of evaluations from Single-Eval and AIME for LeetCodeHard problem: minimum-number-of-visited-cells-in-a-grid.