# OpenReview forum: "AIME: AI System Optimization via Multiple LLM Evaluators"
_ICLR.cc/2025/Conference — ICLR 2025 Conference Withdrawn Submission_

### Official Review · Reviewer_rVtj · 2024-11-03

**Soundness:** 1
**Presentation:** 2
**Contribution:** 2
**Rating:** 3
**Confidence:** 4

**Summary:**

This paper claims that when evaluating an AI-generated output in the context of text-based optimization, AI system optimization via multiple LLM evaluators (AIME) is preferred over a single LLM evaluator. The paper formulated the iterative text-based optimization problem and shows that the suboptimality of a linearly combined evaluator is bounded by the corresponding mixture distribution of the basis evaluators. The paper empirically shows that AIME significantly outperforms the single LLM evaluator approach in error detection and optimization gain on the LeetCodeHard and HumanEval datasets. Ablation studies are performed to demonstrate the impact of the combination of evaluators.

**Strengths:**

1. The proposed approach is novel.

**Weaknesses:**

1. The additive nature of the evaluations is not clearly stated in section 2.
2. The additivity assumption on the aggregation function, which I believe is the key to proving the theory, is not stated in the theorem statement nor anywhere in section 3 before it is used on line 210.
3. The proof of theorem one should include the case of more than two evaluators.
4. In section 3.2, the representation of evaluation is changed from addable to concatable without a clear motivation. This makes the paper inconsistent.
5. The paper proposes multiple evaluators for general-purpose evaluation. However, the experiments only validate it on instance optimization for code generation.
6. The error detection measure evaluated in section 4.1 seems only to study the case when there are failed test cases. The aspect that a good evaluator should also be accurate when there is no error is not tested.
7. The scope claimed in the title is too broad. I recommend replaying "AI System Optimization" with "Text-based Optimization" or something similar.
8. There are mistakes in the writing, i.e., "cite ipm" in the parenthesis on line 158.

**Questions:**

1. Is the evaluation considered as scalers, strings, or something else? Are they assumed to be different types in 3.1 and 3.2?
2. Is that true theorem 1 only applies to the case when the mixture constants $\alpha_k$s are the same for the mixture evaluator distribution and the linearly combined evaluator $g(\cdot)$?

---

### Official Review · Reviewer_9Z5v · 2024-11-03

**Soundness:** 1
**Presentation:** 2
**Contribution:** 2
**Rating:** 3
**Confidence:** 4

**Summary:**

The paper studies the problem of optimizing LLMs using feedback in an iterative manner. Using other LLMs as a proxy for the evaluators, the proposed method recommends the use of *multiple* LLMs instead of a single LLM as is common in practice. Along with a rigorous empirical evaluation of this proposal, the paper also provides theoretical justification for this approach under some idealized assumptions.

**Strengths:**

- Expansive empirical evaluation with insightful ablations.
- That having diversity of roles for evaluation helps is a useful takeaway, much like it has been shown to help for training [1].

---
[1] Rame, Alexandre, et al. "Rewarded soups: towards pareto-optimal alignment by interpolating weights fine-tuned on diverse rewards." Advances in Neural Information Processing Systems 36 (2024).

**Weaknesses:**

- The idea of using multiple LLMs as evaluators has been proposed in some prior works [1,2]. Moreover, the theoretical justification is very idealized and disconnected from practical setups. This diminishes the overall contribution and the work is expected to have a heavy emphasis on its empirical contribution (more on this next).
- Theory:
  - The evaluation suboptimality bounds are for the case of one-shot evaluation rather than iterative evaluation and improvement, the latter being the setup of the paper.
  - The proof is strewn with typos and errors. A few examples:
    - Equation (4): Like Equation (2) the leading term should be $|e_\max|$ rather than $|e^*|$.
    - [L204]: The expression for $\Delta_1$ has an undefined $l$. This statement also introduces the above error.
    - Last equation in the equation block for $\Delta_2$ randomly introduces $e^*$.
  - The idea of using multiple evaluators heavily relies on the last statement “as we increase the number of mixture components [...] under certain assumptions.”. The assumptions have not been stated.
- Empirical evaluation:
  - EDR: The definition seems to have some error. $\frac{1}{|Z_\text{fail}|} \sum_{z \in Z_\text{fail}} q_z$ evaluates to 1, since $Z_\text{fail}$ by definition is the “set of iterations with at least one failed test case”. What is the actual definition? And why is this metric so significantly favorable for AIME relative to SR and CR?

Nitpicks:
- The paper is hastily written. This is indicated by typos in the paper, some examples of which are:
  - Third row of Table 3 highlights AIME as being more performant, whereas the numbers for Single-Eval are higher.
  - [L159] has an unaddressed comment “cite ipm”
  - [L185] “combine the losses”: the *evaluations* are combined, not losses.

---
[1] Verga, Pat, et al. "Replacing Judges with Juries: Evaluating LLM Generations with a Panel of Diverse Models." arXiv preprint arXiv:2404.18796 (2024).

[2] Kim, Seungone, et al. "The BiGGen Bench: A Principled Benchmark for Fine-grained Evaluation of Language Models with Language Models." arXiv preprint arXiv:2406.05761 (2024).

**Questions:**

- Across the two benchmarks, why is the amount of performance improvement with AIME so significantly different (for example, Figures 2 and 4)?

---

### Official Review · Reviewer_5ZXW · 2024-11-04

**Soundness:** 4
**Presentation:** 3
**Contribution:** 2
**Rating:** 3
**Confidence:** 3

**Summary:**

The authors present an ensemble-based system for prompt optimization in agentic systems. This is built by looking at the kinds of errors made by individual LLM evaluators. They conduct a detailed study showing the strength of their system.

**Strengths:**

- The paper is decently well-written and reads like a paper one would expect to see at ICLR.
- The evaluation is sufficiently extensive and shows strong performance over the baseline.
- The approach is quite modular, very practical, and easily extendable.
- I haven't seen a paper exposing exactly this result before.

**Weaknesses:**

- My biggest issue here is in the strength of the contribution. We know that combining several models will improve performance (i.e., ensemble methods). We know this holds true with LLMs as well when solving general tasks (Schmidhuber's Mindstorms paper to name just one). So what exactly is the new thing this work says? Is it just what roles the evaluators should take on to get good performance?
- I would have expected that there would be a fairly extensive discussion of ensemble methods in this paper and that it would be framed as one.
- The Rumelhard citation for backprop should be accompanied by the earlier work that preceded it.
- Less easy to address, I feel like there's something missing on top of the evaluators. Without that, it just feels like another paper saying ensembles work.

The above is my reading of this work. However, the reviewing process is noisy, and no matter how anyone feels about their evaluation, the data says that even the best reviewers can only provide truly medium-confidence reviews. Thus, I'm likely to acquiesce and change my score by a non-trivial amount if the other reviews disagree or the authors can correctly do away with my concerns.

**Questions:**

Please respond to the above when possible as well.

- What does section 3.1 really say? How is this different to all the proofs that exist for ensembles?
- Figure 3 makes me think of chain-of-thought. Did you consider looking at the effect of that on the performances here?
- In Figure 4, why is the EDR so different? This seems like the defining difference between this work and the ensemble/the work mentioned above (and the thousand similar ones out there).

---

### Official Review · Reviewer_xcee · 2024-11-04

**Soundness:** 2
**Presentation:** 2
**Contribution:** 2
**Rating:** 5
**Confidence:** 4

**Summary:**

This paper proposes an alternative to evaluating LLMs along multiple axes by specifying different criteria such as logic, correctness, clarity, etc. The primary metric is the proportion of errors detected using their approach compared with a baseline in which all criteria are mentioned and evaluated in a single shot. The proposed mechanism, AIME, achieves higher error detection rates compared to the baselines and comparable or higher success rate than the baselines on the HumanEval and LeetCodeHard datasets.

**Strengths:**

- The method proposed is general, conceptually simple, and could also be applied to other domains where multiple objectives need to be optimised.
- Figure 1 clearly shows the approach and its advantages over a single LLM evaluation and clearly motivates the work.
- Most of the results show non-trivial improvements over the baselines considered.
- The role-wise analysis in Table 1 gives a nice understanding of the contribution of each role in determining performance.

**Weaknesses:**

- An important baseline that is missing and could potentially give superior results compared to the current baselines is: a single prompt containing descriptions of all criteria, as is the case with SingleEval, but with $k$ independent responses sampled from the LLM with temperature $> 0$. This would arguably give a much fairer comparison than allowing more tokens (3600 for SingleEval versus 3600/$k$ for AIME) since literature in LLMs contains evidence of majority voting or sampling methods outperforming greedy response selection [1, 2]. Furthermore, the results shown in Figure 5 clearly motivate this baseline: increasing the number of evaluators is shown to be helpful even when the same role is maintained.
- There is scope for improving the writing, especially Section 3.2, which contains the overview. It would also be helpful to have a clear description of the roles or criteria in one place and then discuss which ones are more consequential in obtaining the gain in performance (Section 4.3.2 mentions that readability, runtime, and code redundancy evaluators are secondary and leads to worse performance than the SingleEval baseline).
- There are some readability issues: the font size in the plots in Figure 4 is too small to be legible.
- From the results it seems that HumanEval is perhaps not the best dataset to demonstrate the strength of AIME; the performance in terms of success rate and completion rate for all methods is comparable, and the gains seem to be within noise (Tables 3 and 4 in Appendix).
- There is a claim made in the paper: "AIME is more thorough", which seems to be more of a qualitative observation than a quantitative result. It would be better if the paper were to clarify why longer responses are considered more thorough. The example shown in Figure 3 is clear enough, but the examples shown in the appendix seem to indicate that longer answers are considered to be more thorough, which may not always be the case. It would be better to substantiate the argument with what constitutes thoroughness in responses (whether it is the length of the response or something else).

[1] Learning to summarize from human feedback; Stiennon et al, 2020
[2] Self-Consistency Improves Chain of Thought Reasoning in Language Models; Wang et al, 2023.

**Questions:**

- The results in Table 2 in the Appendix seem to indicate that for LeetCodeHard, the method is mostly invariant to temperature, whereas the difference in sampled responses for HumanEval seems more prominent. Could you explain why that is the case, perhaps with some qualitative examples of generations?
- Given the error detection phrases, if the LLM were to output something like "This code is correct, it has to be incorrect because it does not pass the first test", would that count as 2 errors or one? In other words, is there any deduplication done while computing the EDR and other metrics?
- There are 2 temperature settings mentioned: temperature = 0 in the LLM Setup Details in Section 4, and temperature = 1 in the Robustness to Adversarial Evaluator paragraph in Section 4.1. Could these choices be explained and also clarified in the main text?
- The SingleEval performance on HumanEval (Figure 2, left) is very low; could the authors provide some examples of generated output that lead to such a low value?

---

### Note · Authors · 2024-12-04

**Comment:**

We thank the reviewers for the insightful feedback, which will help improve the paper.

Regards,

Authors

**Withdrawal Confirmation:**

I have read and agree with the venue's withdrawal policy on behalf of myself and my co-authors.